

# Modeling Lake Titicaca Daily and Monthly Evaporation

Ramiro Pillco Zolá[1], Lars Bengtsson[2], Ronny Berndtsson[2], Belen Martí-Cardona[3], Frederic Satgé[4], Franck Timouk[5], Marie-Paule Bonnet[6], Luis Mollericon[1], Cesar Gamarra[7], José Pasapera[7],

[1] Instituto de Hidráulica e Hidrología, Universidad Mayor de San Andrés, La Paz, Bolivia

[2] Division of Water Resources Engineering and Center for Middle Eastern Studies, Lund University, Lund, Sweden

[3] Department of Civil and Environmental Eng., University of Surrey, Guildford, UK

[4] CNES, UMR HydroSciences, Univeristy of Montpellier, Place E. Bataillon, 34395 Montpellier Cedex 5, France

[5] IRD, UMR5563); Obs. Midi-Pyrénées, Université P. Sabatier, Toulouse, France

[6] IRD, UMR Espace-Dev, Maison de la télédétection, 500 rue JF Breton, 34093 Montpellier cedex 5, France

[7] IMARPE, Puno, Perú

*Correspondence to*: R. Pillco Zolá (rpillco@umsa.edu.bo)

**Abstract.** Lake Titicaca is an important water ecosystem of South America. Due to uncertainties in estimating the evaporation losses from the lake, surface water storage calculations are uncertain. In this paper, we try to improve

evaporation loss estimations by comparing different methods to calculate daily and monthly evaporation from Lake Titicaca. These were: water balance, heat balance, mass transfer method, and the Penman equation. The evaporation was computed at daily time step and compared with estimated evaporation using mean monthly meteorological observations. We found that the most reliable method of determining the annual lake evaporation is using the heat balance approach. To estimate the monthly lake evaporation using heat balance, the heat storage changes must be known in advance. Since convection from the

surface layer is intense during nights resulting in a well-mixed top layer every morning, it is possible to determine the change of heat storage from the measured morning surface temperature. The mean annual lake evaporation was found to be 1700 mm. Monthly evaporation computed using daily data and monthly means resulted in minor differences.

Key words: Lake Titicaca, heat balance, water balance, lake evaporation.

## 1 Introduction

Lake Titicaca, the largest freshwater lake in South America, is located in the endorheic Andean mountain range plateau – Altiplano, straddling the border between Perú and Bolivia (Fig. 1). The lake plays an essential role in shaping the semiarid Altiplano climate and feeds the downstream Desaguadero River and Lake Poopó (Pillco & Bengtsson, 2006; Abarca-del-Río et al., 2012); thus supplying the inhabitants with water resources for domestic, agricultural, and industrial use (Revollo, 2001). Anthropogenic pressure on the Altiplano water resources has increased during the last decades due to population

growth and increased irrigated land (FAO, 2005; Canedo et al., 2016; Satgé et al., 2017), as well as due to industrial pollution (UNEP, 1996; CMLT, 2014). Revollo (2001) pointed out that the maximum surplus of the Lake Titicaca to supply downstream users was dramatically lower than the estimated demand. The challenge of managing water resources in the



Altiplano Basin is further exacerbated by climate conditions. Annual rainfall can fluctuate up to 50% (Garreaud et al., 2003), while global warming is expected to intensify evaporation losses. The combined impact of these pressures becomes evident at the downstream end of the system, where Poopó Lake is situated. In recent years this lake suffered extreme water shortages, including its complete drying out in December 2015 (Satgé et al., 2017).

5   Lake Titicaca has a large surface area of about 8500 km$^2$ on average. Over a certain water surface level, the lake spills at the South East end and feeds the Desaguadero River. However, the major water output from Lake Titicaca is due to evaporation, which accounts for more than 90% of the total water output (Roche et al., 1992; Pouyaud, 1993; Talbi et al., 1999; Delclaux et al., 2007). In recent years, Lake Titicaca's level dropped below the outlet threshold for some periods. Thus, a small increase in evaporation or decrease in precipitation may turn the lake into a closed system with no outflow.

10  It is essential to improve the knowledge of the lake evaporation since evaporation dominates the water balance. This is especially important in light of anthropogenic pressure and climate change. Previous studies of Lake Titicaca evaporation have all been based on monthly meteorological observations. Due to the importance of lake evaporation, detailed calculations using daily as well as monthly observations may be necessary. Consequently, this paper investigates different methods for calculating evaporation using both daily and monthly data.



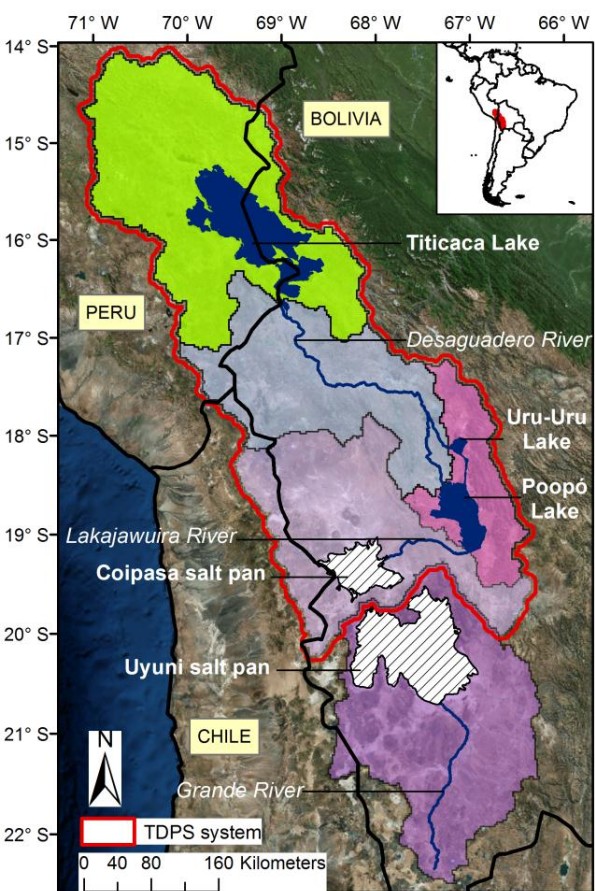

**Figure 1: Lake Titicaca and the TDPS system within the Altiplano**

## 1.1 Previous investigations

In one of the first lake evaporation studies for Lake Titicaca, observations during 1956-1973 were used (Carmouze et al., 1977). By applying the water balance method, they estimated the Lake Titicaca mean annual lake evaporation equal to 1550 mm year-1. Taylor and Aquize (1984) applied a bulk transfer approach for a shorter period and determined the lake evaporation to be 1350 mm year-1. The largest reported annual evaporation is from using the energy balance approach. Richerson et al. (1977) found the lake evaporation equal to 1900 mm year-1. Later, Carmouze (1992) used the same approach and found the lake evaporation to be 1720 mm year-1. Using observations for the period 1965-1983 and the water balance method, Pouyaud et al. (1993) found the mean annual evaporation equal to about 1600 mm year-1. Thus, the mean annual evaporation has been estimated in the range 1350-1900 mm year-1. While the precipitation can vary much from year to year, the large range of calculated annual evaporation, 1350 to 1900 mm, is not likely to be true. Rather, the large range is due to uncertainties in the methods used for estimating the evaporation. Recently, Delclaux et al. (2007) studied the





evaporation from Lake Titicaca using in-situ pan evaporation measurements, energy balance, mass transfer, and the Penman methods. They concluded that the mean annual evaporation may be about 1650 mm year-1, with low seasonal variation between 135 mm in July (winter) and 165 mm in November (summer).

## 2 Study area

5    Lake Titicaca is a unique biosphere due to its large depth and volume with tropical climate conditions situated at high altitude. It is located in the northern part of the Perú-Bolivian Altiplano, between latitude of 15° 25' and 16° 35' south. It is surrounded by the eastern and western Andean mountains. The total Lake Titicaca basin area including the lake itself is close to 57000 km$^2$ with a mean elevation higher than 4000 m a.m.s.l. The lake surface is 8560 km$^2$ at an elevation of 3810 m a.m.s.l. The outlet sill is at 3807 m a.m.s.l. The lake volume is about 903 km$^3$ with a corresponding mean depth of 105 m

10   (Boulang'e and Aquize, 1981; Wirrmann, 1992). The only outlet is the Desaguadero River that ends in the shallow Poopó Lake. The modern Lake Titicaca consists of the major Titicaca (Lake Chuquito), which is 284 m at the deepest point, and the smaller Titicaca (Lake Huiñamarca). The latter lake represents 1200 km$^2$ with a maximum depth of 35 m below spill-level. The threshold between the two lake basins is at 19 m below spill-level (see Fig. 2). The lake is described by Dejoux and Iltis (1992) and in the Encyclopedia of Lakes and Reservoirs edited by Bengtsson et al. (2012).

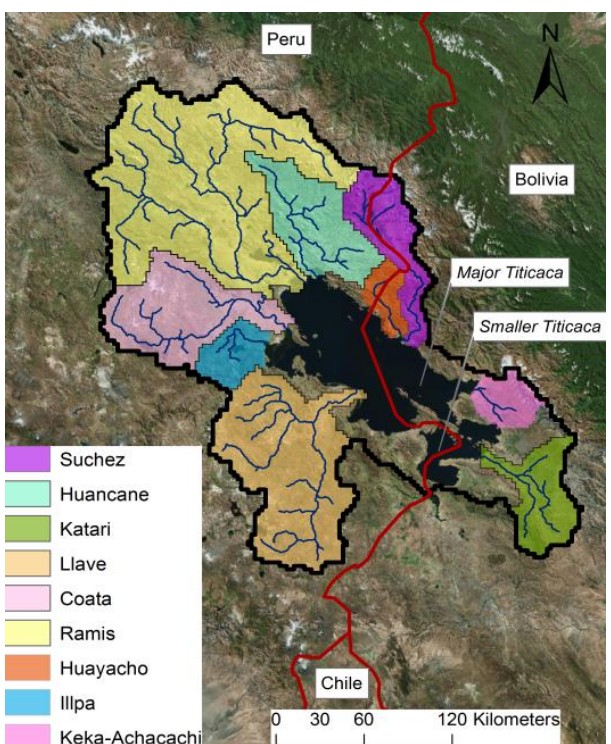

**Figure 2: Lake Titicaca Basin with sub-basins and major and smaller lakes.**



## 2.1 Hydrology

The Lake Titicaca watershed is a part of the TDPS system (Titicaca, Desaguadero, Poopó, and Salares) within the Altiplano (Revollo, 2001). The Poopó Lake is considered a terminal lake with only one discharge event into the downstream Coipasa salt pan occurring in the last century (Pillco & Bengtsson, 2006). The Lake Titicaca Basin itself includes the sub-basins Katari, Coata, Huancane, Huaycho, Ilave, Illpa, Keka-Achacachi, Ramis, and Suchez. The largest is the Ramis River Basin with an area of 15000 km$^2$, representing 30% of the total upstream basin (Fig. 2). The mean flow of the Ramis River for the period 1965-2011 was 72 m$^3$ s$^{-1}$. The mean outflow from Lake Titicaca through the Desaguadero River, for the same period was 35 m$^3$ s$^{-1}$. During these 50 years, Lake Titicaca experienced large changes in water level, with a mean close to 3808.1 m a.m.s.l., which is about 1 m above the outlet threshold (Pillco & Bengtsson, 2006). From low to high water level, the Lake Titicaca water surface area might change from a minimum of 7000 to a maximum of 9000 km$^2$.

## 2.1 Climate

The northern part of the Altiplano is semiarid, while the southern part including the biggest salt pans in the world is arid (TDPS, 1993). The climate is further characterized by a short wet season (Dec-Mar) and a long dry season (Apr-Nov) (Garreaud et al., 2003). The average precipitation over the Lake Titicaca Basin is about 800 mm year$^{-1}$, out of which more than 70% fall during the wet season (Garreaud et al., 2003). Over the lake, annual precipitation is assumed to vary from 1200 mm year$^{-1}$ in the central part to 800 mm year$^{-1}$ along the shores (TDPS, 1993). January is the wettest (about 180 mm) and July the driest (less than 10 mm) month. In the central and southern parts of the Altiplano, the total annual precipitation is about 350 mm (Roche et al. 1992; Pillco & Bengtsson, 2006) and less than 200 mm over the Salares in the southernmost areas (Satgé et al., 2016). The seasonal variability of precipitation in the basin is related to changes in the upper troposphere circulation. During the Austral summer, an upper-level cyclone is established southeast of the central Andes. The Bolivian High brings easterly winds and allows influx of moisture from the central continent over the plateau during periods intensifying the precipitation (Garreaud, 1999; Vuille et al., 2000).

The daily air temperature over Lake Titicaca is rather constant throughout the year, usually varying between 7 and 11oC but sometimes up to 20oC in summer. The Titicaca region is more humid than the more southern parts of the Altiplano. The relative humidity varies between 52 to 68% as monthly average with diurnal variation between 33 and 80%. According to Carmouze (1992), the dominant wind on the lake is north-west to south-east direction, with mean monthly wind velocity close to 2 m s$^{-1}$, rarely reaching 5 m s$^{-1}$ at daily time step. The general climate and hydrology are summarized in Table 1. The total river inflow was estimated through a representative area approach based on the Ramis River discharge.



**Table 1. Climatological and hydrological components of Titicaca Lake for 1966-2011**

| Months | Ramis flow ($m^3/s$) | Inflow ($m^3/s$) | Outflow ($m^3/s$) | Lake depth (m) | Precipitation (mm/month) | Air temperature ($^o$C) | Relative humidity (%) | Wind velocity (m/s) |
|---|---|---|---|---|---|---|---|---|
| **Jan** | 173.5 | 549.0 | 31.3 | 283.9 | 177.0 | 10.2 | 67.5 | 1.9 |
| **Feb** | 198.8 | 629.4 | 39.4 | 284.1 | 141.6 | 10.1 | 67.4 | 1.8 |
| **Mar** | 190.4 | 602.8 | 49.8 | 284.4 | 126.8 | 10.0 | 67.3 | 1.8 |
| **Apr** | 104.9 | 332.1 | 50.9 | 284.5 | 50.6 | 9.8 | 61.9 | 1.7 |
| **May** | 38.7 | 122.6 | 46.5 | 284.5 | 13.2 | 8.8 | 54.9 | 1.7 |
| **Jun** | 20.7 | 65.5 | 41.8 | 284.4 | 7.3 | 7.8 | 52.9 | 1.9 |
| **July** | 14.5 | 46.0 | 37.0 | 284.3 | 6.6 | 7.7 | 52.8 | 1.9 |
| **Aug** | 11.0 | 34.9 | 32.1 | 284.2 | 13.2 | 8.3 | 53.7 | 2.0 |
| **Sep** | 9.9 | 31.3 | 28.1 | 284.1 | 29.9 | 9.2 | 55.1 | 2.1 |
| **Oct** | 14.0 | 44.2 | 24.1 | 284.0 | 45.5 | 10.2 | 55.4 | 2.1 |
| **Nov** | 24.0 | 76.0 | 21.7 | 283.9 | 56.7 | 10.8 | 56.4 | 2.1 |
| **Dec** | 55.2 | 174.7 | 22.1 | 283.9 | 102.5 | 10.6 | 62.4 | 2.0 |

## 3 Methods

### 3.1 Theoretical background

The Lake Titicaca surface water is cold with a temperature that remains 12-17 $^o$C throughout the year, and below 40 m depth the temperature is almost constant (Richerson et al., 1977). The water is usually warmer than the air, which means that the air above the lake water is unstable. The air temperature shows large diurnal variations, some days it exceeds 15 $^o$C in summer. At an average terrain elevation above 4000 m the solar radiation is strong and the atmospheric pressure is low, which means that the ratio between sensible and latent heat flux (Bowen ratio) is lower than at sea level. Thus, the conditions for evaporation from Lake Titicaca are thus particular.

From the several methods used for estimating Titicaca Lake evaporation, provided that all inflows to the lake including precipitation on the lake surface are observed, the most reliable method to calculate evaporation over a year is probably the water balance method. However, all river inflows to Lake Titicaca are not measured. Further, the lake is so large that the precipitation measured at shore is likely to be different from that on the lake surface. To determine the evaporation rate using the aerodynamic mass transfer approach, the atmospheric vapor pressure and surface temperature must be known. Furthermore, a wind function must be used because the atmosphere over Lake Titicaca is unstable most of the time. This means that the wind function may be different from the function used for most other lakes.



It can generally be assumed that during a year the lake water temperature returns to the value at the beginning of the year. Thus, for the heat balance, it is sufficient to know the annual net radiation, provided that the sensible heat flux can be estimated from the constant Bowen ratio. When using the method for shorter time periods, the time variation of the lake water temperature profile must be known. The heat balance approach and the aerodynamic method can be combined. The

5 Penman method is such a combined approach (Penman, 1948). A wind function must be included also in this approach.

## 3.2 Lake evaporation models

Four evaporation estimation methods were applied in this study, water balance, energy balance, mass transfer, and the Penman method. These approaches have previously been used by other researchers to estimate Lake Titicaca evaporation at a monthly time step (Carmouze, 1992; Garcia, 2004; Pouyaud, 1993; Delclaux et al., 2007).

### 3.2.1 Energy balance approach

The energy balance is:

$$\lambda E = \frac{R_n - Q_{heat}}{1 + \beta} \tag{1}$$

where $\lambda$ is the latent heat vaporization, $E$ is evaporation rate, $R_n$ is net radiation, $Q_{heat}$ is heat storage within the water, and $\beta$ is the Bowen ratio:

$$\beta = \gamma \frac{T_w - T_a}{e_w - e_a} \tag{2}$$

$$\gamma = \frac{c_p p_a}{0.622\lambda} \tag{3}$$

where $\gamma$ is the psychrometric constant, $T_w$ and $T_a$ are the surface water and air temperature, respectively, $e_w$ and $e_a$ are the water surface and air vapor pressures, respectively, $c_p$ is the specific heat capacity, and $p_a$ is the atmospheric pressure. The psychrometric constant and thus, also the Bowen ratio, are lower at this high altitude than at sea level. The net radiation is

20 the sum of net short-wave and net long-wave radiation. The net short wave radiation is $R_s$ (1- albedo), where $R_s$ is the solar radiation reaching the lake. The atmospheric long-wave radiation as well as the back radiation are computed from the Stefan law. The emissivity of the water is well known, it was set to 0.98 and the emissivity atmosphere must be known. The emissivity of the atmosphere depends on humidity, temperature, and cloudiness. Atmospheric emissivity accounting for clouds was proposed by Crawford and Duchon (1999):

$$\varepsilon_e = (1 - s) + s\varepsilon_o(T_a, e_a) \tag{4}$$

$$s = \frac{R_s}{R_{s,o}} \tag{5}$$



$$R_{s,0} = R_a e^{\left(\frac{-0.0018P}{K_t \sin \phi_{24}}\right)} \qquad (6)$$

$$\varepsilon_o = 1.18 \left(\frac{e_a}{T_a}\right)^{\frac{1}{7}} \qquad (7)$$

where $s$ is the proxy cloudiness defined as the ratio of measured incoming solar radiation ($R_s$) to the solar radiation received for clear sky conditions ($R_{s,o}$), $\varepsilon_0$ is the emissivity in clear-sky condition, which is determined from the vapor pressure $e_a$ expressed in hPa and $T_a$ temperature in Kelvin. The $\Phi_{24}$ is mean daily sun angle. The constant 1.18 describes the attenuation defined for the region according to Lhome $et$ $al.$ (2007). The extraterrestrial solar radiation ($R_a$) is determined as function of local latitude and altitude, and time of year using the turbidity coefficient $K_t$=0.85.

The energy equation is fairly easy to use over a full year, since the lake water usually returns to its initial state when computations were started, or when $Q_{heat}$ equals zero. When using the approach over shorter periods the variation of the water temperature in the lake must be accounted for. In Eq. (1), the change of heat storage is included. From temperature water profiling observations, it was assumed that the water temperature below 40 m does not change from month to month. The temperature, $T_w$, above this mixing depth, $h_{mix}$, changes but remains almost homothermal after convective mixing during night (Richerson et al., 1977), which also is corroborated by our own field investigations. Thus, the change of heat content can be estimated from measured surface temperature:

$$Q_{heat} = \rho c_p \frac{V_{mix}}{A_{lake}} \frac{\partial T_w}{\partial t} \qquad (8)$$

where $\rho$ is density of water, $c_p$ is the specific heat capacity of water, and $V_{mix}$ is the volume above the mixing depth. Carmouze et al. (1992) introduced the concept of exchange of heat between surface and deep water using the energy balance concept. The results of Carmouze $et$ $al.$ (1992) were compared to the calculation results in the present study.

**3.2.2 Mass transfer approach**

The mass-transfer aerodynamic approach is used in various models based on Dalton's law. The latent heat transfer is related to the vapor pressure deficit. Most often a linear wind function is used (e.g., Carmouze et al., 1992)

$$E = (a + bU)(e_w - e_a) \qquad (9)$$

where $E$ is evaporation rate (mm/day), $U$ is wind velocity (m/s), ($e_w$ - $e_a$) is vapor pressure deficit (mbar). The parameter $a$ accounts for unstable atmospheric conditions. Carmouze $et$ $al.$ (1992) used $a$=0.17 mm mbar[-1] day[-1] and $b$=0.30 mm mbar[-1] s m[-1]. These coefficients are higher than found in several Russian studies, for example Kuzmin (1961). These coefficients are higher than those determined in several studies undertaken in Russia, such as Kuzmin (1961). These last studies showed coefficients to be $a$=0.13 and $b$=0.06 (with units as above). A possible explanation for the high values used by Carmouze et al. (1992) may be that the wind was measured at a rather wind protected location. Another explanation may be that the wind speed was low as affected by surface roughness.





### 3.2.3 Penman approach

The Penman equation is a combination of energy balance and mass transfer for evaluating open water evaporation:

$$E = \frac{\Delta}{\Delta + \gamma}\frac{(R_n - Q_{heat})}{\lambda \rho} + \frac{\gamma}{\Delta + \gamma}(a' + b'U)(e_s - e_a) \tag{10}$$

where $E$ is open water evaporation. The slope of the water pressure-temperature curve is denoted by $\Delta$, $e_s$-$e_a$ is the saturation

deficit of the air, here $e_a$ is $(1\text{-}RH)\,e_s$. Delclaux et al. (2007) applied the Penman equation to Lake Titicaca using $a' = 0.26$ and $b' = 0.14$ after optimizing (with units as above), which again is higher than usually reported. These coefficients need not to be the same as in the mass transfer approach.

### 3.2.4 Lake water balance model

The water balance approach was applied to calculate water levels in Lake Titicaca in a previous study by Pillco & Bengtsson

(2007). The water balance is:

$$A_{lake}\frac{\partial h}{\partial t} = (P - E)A_{lake} + Q_{in} - Q_{out} \tag{11}$$

where $\partial h/\partial t$ represents change in water depth, $P$ is precipitation on the lake, $E$ is evaporation from open water, $A_{lake}$ is water surface of the lake, which is a function of depth, $Q_{in}$ is inflow to the lake, and $Q_{out}$ represents the outflow from the lake. The area-depth relationship as found by Pillco & Bengtsson (2007) was used. Computations were carried out at a monthly time

scale for the period 1966-2011. As already pointed out, the most reliable method of computing evaporation over long periods is probably the water balance method. However, the inflow to Lake Titicaca is not measured in all rivers. Furthermore, the lake is so large that the precipitation measured on shore is likely to be different from that on the lake. Also, because of wind effects, the measured lake level at one station may not be representative for the entire lake.

### 3.2.5 Possible errors when using monthly averages

The evaporation during individual days is not important for the water balance but only over longer periods as months. However, since the equations for calculating evaporation are not linear, the monthly evaporation computed from monthly mean meteorological data may differ from what is found when data with higher time resolution are used. In the aerodynamic approach the wind speed is multiplied by the vapor deficit. The energy balance approach includes the Bowen ratio, which may differ from day to day and can even be negative for certain periods. If high atmospheric vapor pressure is related to

strong winds, the aerodynamic equation using monthly means can yield lower evaporation estimates than when daily values are used. This is further discussed below. The Bowen ratio changes during a month. When the net radiation is large, the air temperature is likely to be rather high but not necessarily related to high vapor pressure. For this situation, the Bowen ratio is relatively high and the computed evaporation higher than it would have been using a constant monthly Bowen ratio. This means that using monthly averages, the computed evaporation will tend to be low.



## 4 Instrumentation and data

For this study, measurements of hydrological, meteorological parameters and water surface temperature were done on the Lake Titicaca shoreline during 24 consecutive months (2015-2016) and periodically for the lake temperature profile. Observations were taken at 15 min intervals. These records were averaged to daily and monthly values. A Campbell Scientific research-grade automatic weather station (AWS) was installed at the Isla de Luna (latitude 16o01'59'', longitude 69o04'01''), near the Titicaca Lake shore (Fig. 3). The AWS was equipped with a rain gauge sensor, a CS215 probe for measuring relative humidity and air temperature, an A100R vector anemometer and W200P wind vane to measure wind speed, and a pyranometer Skye SP1100 for solar radiation measurement. The surface water temperature was taken from Juli-Puno (latitude 16o12'58'', longitude 69o27'31'') at a distance of 42 km from Isla de Luna. A hand-held thermometer was used to measure water surface temperature at 8-hour intervals.

Hydrological data, such as inflow to the lake were observed at the outlet of Ramis River. The outflow through Desaguadero River was observed at Aguallamaya. This is 40 km downstream of the lake outlet. However, there are only few tributaries between the lake and this point that may contribute to the data uncertainty. The water level was observed at Huatajata at daily time step, shown as depth in Table 1. Additional lake water temperature sounding were carried out close to Isla de Luna (latitude 16o30'00", longitude 69o15'10") for specific days during summer, spring, and winter of the study period, using multiparameter data sond – Hydrolab DS5. The sounding reached a maximum lake depth of 95 m, with water temperature recording each 5 cm at the surface and each 0.5 m below of 1 m depth.

Long-term monthly temperature and wind observations from 1960 onwards were available from the Copacabana weather station mentioned above (Fig. 3; Table 1). The El-Alto station observations, 50 km from Copacabana, were used to fill in 2.5% missing wind data for the period. The monthly precipitation on the lake was determined using the rain gauge at Copacabana and Puno on the lake shore. The total inflow from all rivers was estimated from a representative area approach assuming the specific runoff to be the same for all rivers entering into Lake Titicaca. The long-term outflow from the lake was measured at the outlet of the lake and treated by Molina et al. (2015).





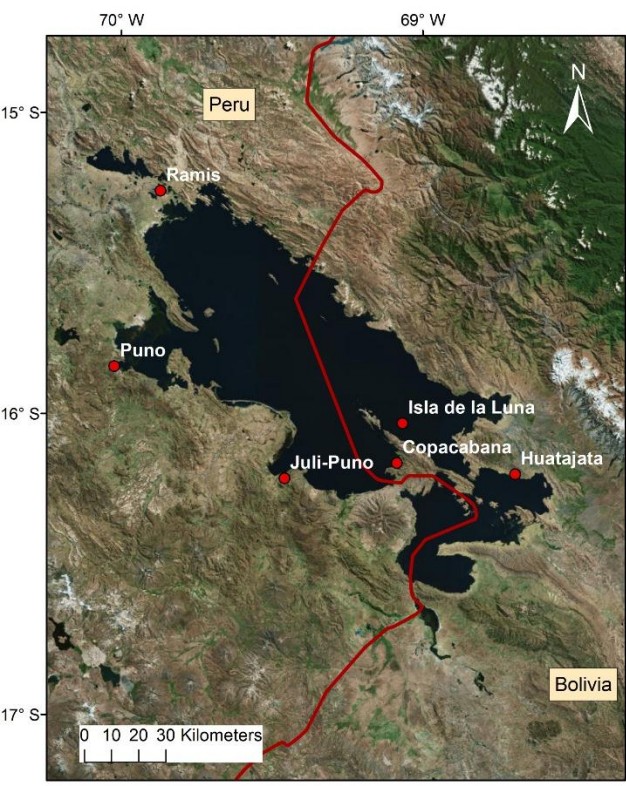

**Figure 3. Location of observation points.**

Table 2 and Table 3 summarize hydrological and meteorological measurements used in this study. Subscripts for vapor pressure are *w* for water, *s* for saturated air vapor pressure, and *a* is for actual air vapor pressure. The computed variables required for evaporation calculations are given in Table 4.



**Table 2. Monthly mean of hydrological variables observed during the 2015-2016 period.**

| Month | Lake depth (m) | Inflow (m³/s) | Outflow (m³/s) | Precipitation (mm/month) |
|---|---|---|---|---|
| **Jan** | 282,9 | 399,8 | 37,7 | 165,4 |
| **Feb** | 283,1 | 639,7 | 53,9 | 146,1 |
| **Mar** | 283,2 | 427,0 | 47,3 | 76,5 |
| **Apr** | 283,3 | 327,9 | 42,1 | 103,9 |
| **May** | 283,3 | 148,4 | 40,2 | 18,1 |
| **Jun** | 283,2 | 80,2 | 32,9 | 7,7 |
| **Jul** | 283,0 | 56,7 | 28,6 | 51,9 |
| **Aug** | 282,9 | 42,6 | 24,4 | 12,5 |
| **Sep** | 282,8 | 36,8 | 20,7 | 26,6 |
| **Oct** | 282,7 | 32,3 | 19,1 | 42,9 |
| **Nov** | 282,6 | 35,7 | 18,2 | 39,0 |
| **Dec** | 282,5 | 92,0 | 18,5 | 87,6 |

5 **Table 3. Monthly averages of main climatic variables observed during the 2015-2016 period.**

| Month | Water surface temp. (°C) | Air temp. (°C) | Wind velocity (m/s) | Ralative humidity (%) | Vapour pressure (mbar) $e_w$ | $e_s$ | $e_a$ | Solar radiation, Rs (W/m²) |
|---|---|---|---|---|---|---|---|---|
| **Jan** | 17.2 | 11.1 | 1.60 | 68.3 | 19.8 | 13.9 | 9.5 | 273.3 |
| **Feb** | 17.3 | 11.4 | 1.61 | 70.3 | 19.9 | 14.2 | 10.0 | 292.4 |
| **Mar** | 17.5 | 12.0 | 1.50 | 66.1 | 20.2 | 14.7 | 9.7 | 273.4 |
| **Apr** | 16.5 | 10.8 | 1.52 | 66.7 | 19.0 | 13.6 | 9.1 | 229.2 |
| **May** | 15.4 | 10.7 | 1.33 | 53.4 | 17.5 | 13.5 | 7.2 | 234.9 |
| **Jun** | 14.3 | 10.1 | 1.30 | 51.8 | 16.7 | 13.0 | 6.7 | 236.7 |
| **Jul** | 13.7 | 9.7 | 1.41 | 50.6 | 16.1 | 12.8 | 6.5 | 237.9 |
| **Aug** | 14.0 | 9.7 | 154 | 54.4 | 16.4 | 12.8 | 7.0 | 265.8 |
| **Sep** | 14.7 | 10.5 | 1.50 | 57.6 | 17.1 | 13.6 | 7.8 | 304.7 |
| **Oct** | 15.5 | 10.9 | 1.63 | 58.2 | 17.6 | 13.9 | 8.1 | 318.7 |
| **Nov** | 16.4 | 11.7 | 1.61 | 57.1 | 18.8 | 14.8 | 8.4 | 332.1 |
| **Dec** | 16.9 | 11.6 | 1.70 | 64.9 | 19.5 | 14.5 | 9.4 | 315.9 |




**Table 4. Monthly average parameters for evaporation calculations for the 2015-2016 period.**

| Month | Atmospheric Emissivity $\varepsilon$ | Bowen ratio $\beta$ | Slope of water vapour pressure $\Delta$ (mbar·100/°C) |
|---|---|---|---|
| Jan | 0.81 | 0.38 | 87.40 |
| Feb | 0.80 | 0.38 | 89.10 |
| Mar | 0.79 | 0.31 | 91.90 |
| Apr | 0.80 | 0.35 | 86.20 |
| May | 0.74 | 0.22 | 85.20 |
| Jun | 0.71 | 0.19 | 82.50 |
| Jul | 0.71 | 0.18 | 80.70 |
| Aug | 0.72 | 0.21 | 80.60 |
| Sep | 0.74 | 0.21 | 84.60 |
| Oct | 0.75 | 0.23 | 86.60 |
| Nov | 0.76 | 0.21 | 90.70 |
| Dec | 0.78 | 0.30 | 90.00 |

Short-wave radiation was measured while long-wave radiation was computed as described above. The average for all
5 components is shown in Fig. 4. The radiation budget is positive every day with a mean of about 150 Wm$^{-2}$ varying from 100
in winter to 200 Wm$^{-2}$ in summer.

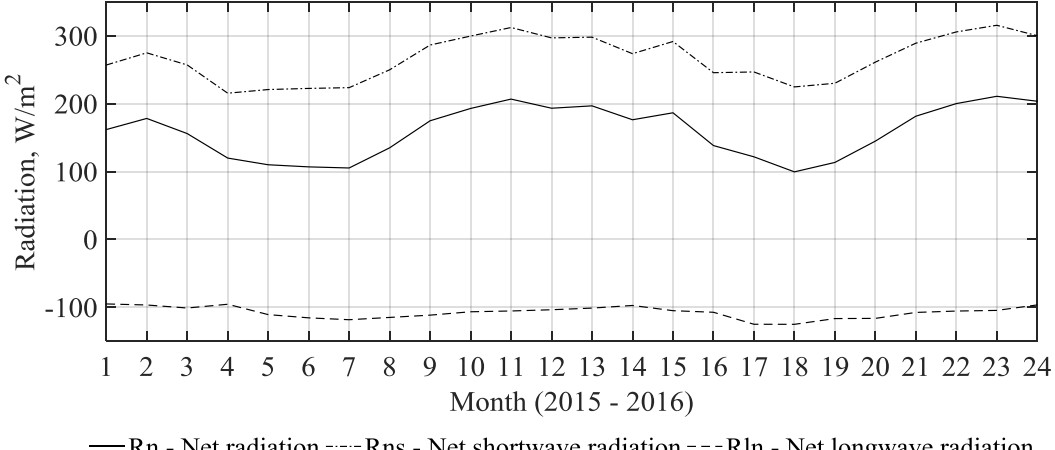

**Figure 4. Monthly average radiative budget for Lake Titicaca 2015-2016.**




## 5 Results and discussion

### 5.1 Monthly data

Detailed energy balance computations over the period 2015-2016 should give good estimates of the total lake evaporation for that period. After 24 months the lake surface temperature at Puno more or less returned to the temperature at the beginning of 2015. Applying this method over the two years gave a total evaporation of 3400 mm, corresponding to a mean annual lake evaporation of 1700 mm year$^{-1}$. When computing the evaporation month by month the change of heat storage was considered in the way previously described. The mixing depth was set to 40 m. The change of the heat storage is shown in Fig. 5. The values suggested by Carmouze et al. (1992) are shown for comparison. The calculated monthly heat storage agrees well with the Carmouze estimates.

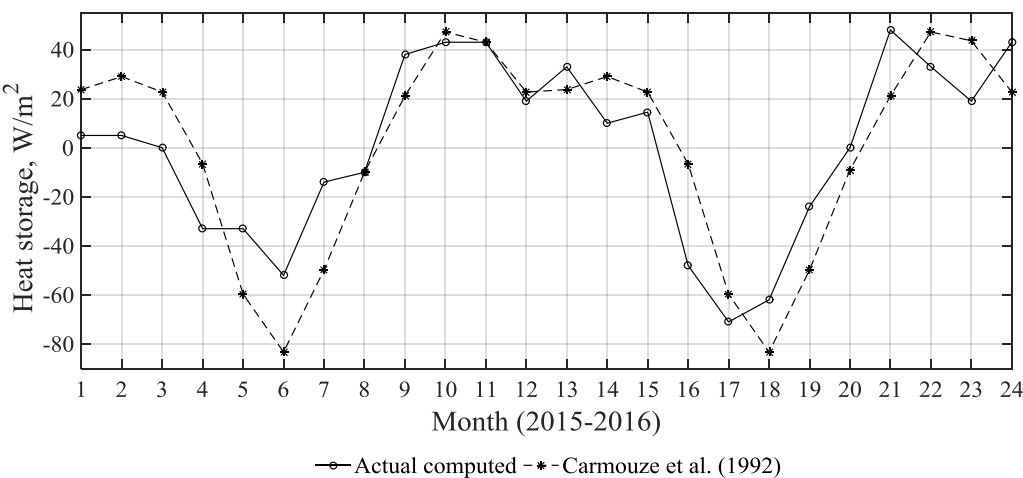

**Figure 5. Change of heat storage during 2015-2016.**

The computed monthly evaporation using monthly average data and the energy balance method was somewhat higher in 2016 than in 2015, 1725 mm as compared to 1680 mm. The evaporation was fairly evenly distributed over the year, being about 140 mm per month, with somewhat lower evaporation rates in July through September (see in Fig. 6).





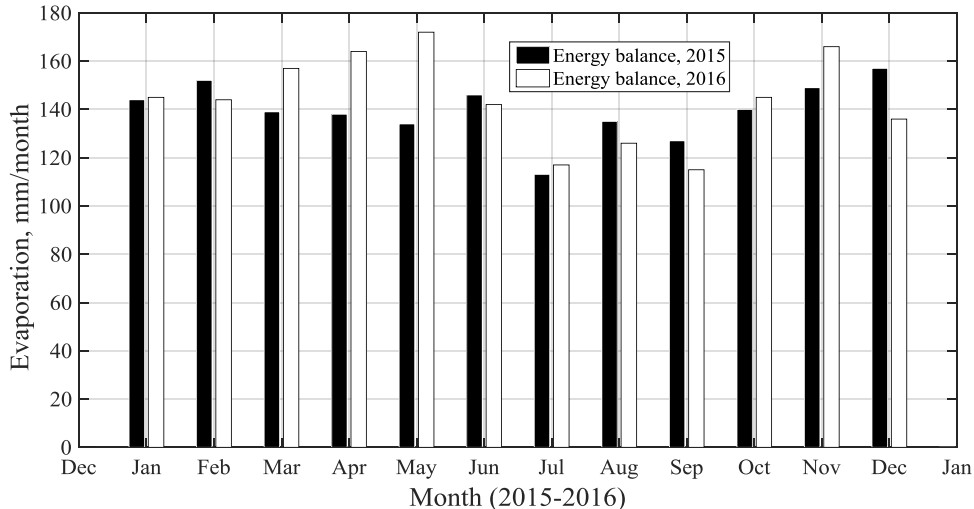

**Figure 6. Monthly evaporation computed using energy balance approach.**

From the energy balance and the water balance methods, the annual evaporation from Lake Titicaca was estimated in the range of about 1700 mm year$^{-1}$. The monthly variation depends on the change of heat storage and therefore the calculated evaporation may be high one month and low the following month. When using the mass transfer approach, similar annual evaporation as from the energy balance approach may be anticipated when applying the approaches over two full years. This may be the case even though there may be differences when comparing monthly calculations. However, when the coefficients suggested by Carmouze et al. (1992) were used the evaporation was much higher than what was found from the energy balance method. A good fit for the total evaporation was found using the coefficients $a$=0.17 mbar, as suggested by Carmouze et al. (1997) and $b$=0.155 mm mbar$^{-1}$ s m$^{-1}$.

The mass transfer computed monthly evaporation over the two years is compared with the energy balance calculations in Fig. 7 for 2015 and in Fig. 8 for 2016. The computed annual evaporation by the last method was 1700 mm in 2015 and 1675 mm in 2016. Consequently, for individual years the two methods gave similar results. This is expected since the coefficients in the mass transfer equation were chosen to fit over the two year period. For individual months, there are deviations. However, it is not possible to note any systematic differences related to different seasons of the year. The largest difference between the two methods for an individual month was about 30 mm.

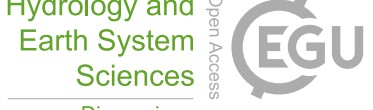

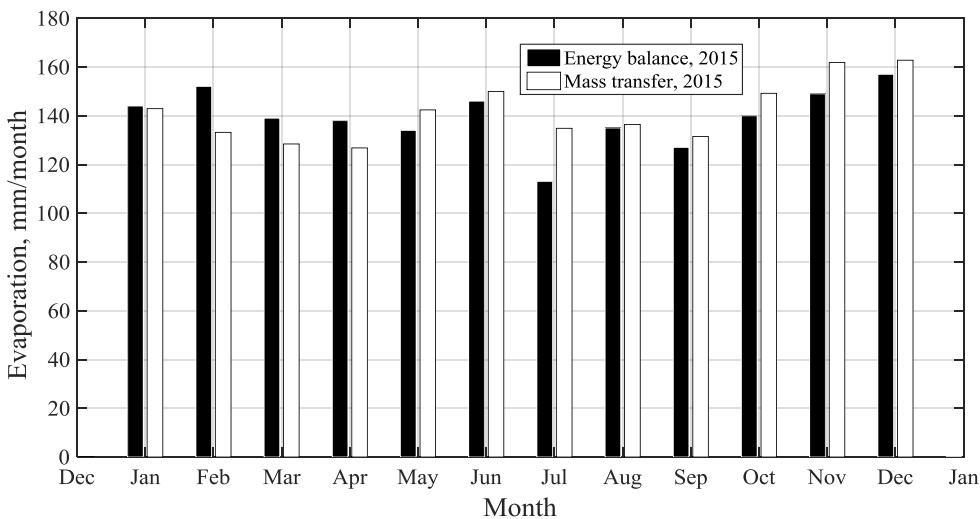

**Figure 7. Comparison of monthly evaporation computed by energy balance and mass transfer method for 2015.**

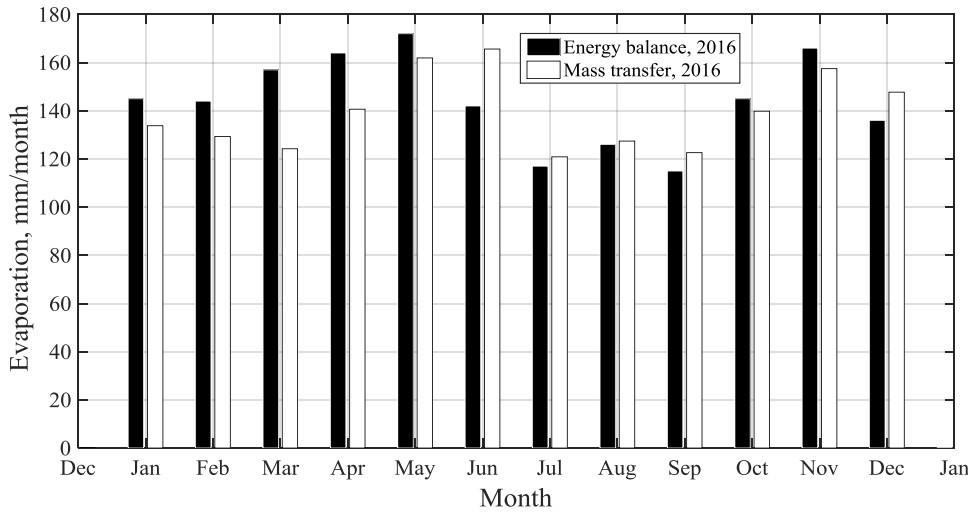

5    **Figure 8. Comparison of monthly evaporation computed by energy balance and mass transfer method for 2016.**

A summary and comparison of all investigated methods for the study period are shown in Fig. 9 and Table 5. As seen from these, the evaporation calculated from water balances differs from the three other methods. The water balance method yielded the largest mean annual and mean monthly evaporation. As well, the same method gave the largest standard deviation. The variation in mean annual evaporation was 1621-1701 mm while it was 1777 mm for the water balance
10   method. Since the lake water level at best can be observed with cm-resolution, individual monthly evaporation becomes highly uncertain. Thus, calculated annual evaporation is better performed using the three other methods. In general, the mass

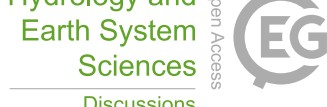


transfer, energy balance, and the Penman method gave similar results and a similar monthly variation as described above while water balances differed significantly, with largest standard deviation and error of mean.

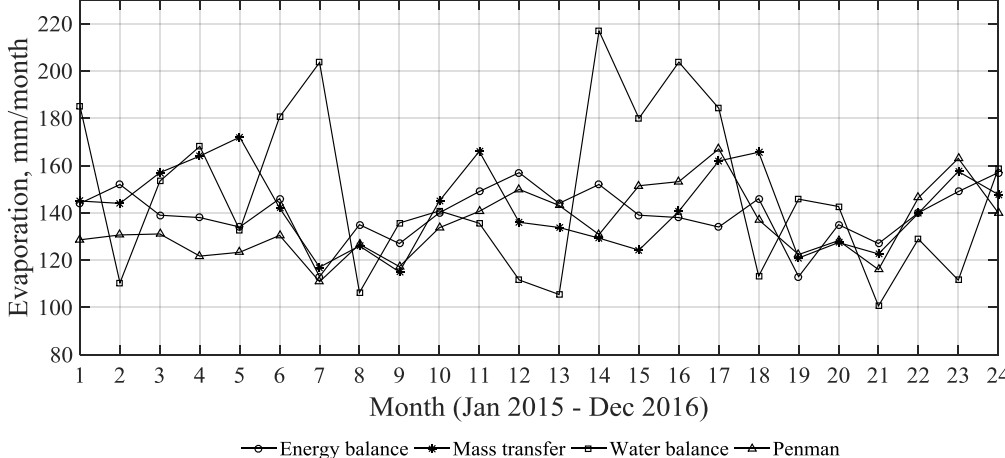

Figure 9. Monthly actual evaporation calculated by the four methods for the period January 2015 to December 2016.

Table 5. Descriptive statistics of monthly actual evaporation calculated by the four methods for the period January 2015 to December 2016.

| Method | Mean annual | Mean monthly | Min | Max | Stand. Dev. | Stand. Error of mean |
|---|---|---|---|---|---|---|
| | (mm/year) | (mm/month) | (mm/month) | (mm/month) | (mm/month) | (mm/month) |
| **Energy balance** | 1701 | 141.8 | 113.0 | 172.0 | 15.4 | 3.1 |
| **Mass transfer** | 1686 | 140.5 | 120.9 | 165.6 | 13.8 | 2.8 |
| **Water balance** | 1777 | 148.2 | 100.6 | 217.0 | 35.1 | 7.2 |
| **Penman** | 1621 | 135.2 | 110.9 | 167.0 | 14.5 | 2.9 |

Water balances were computed as well for the long-term period 1966-2011. During the computation the Alake parameter in the model was assumed constant equal to 8800 km$^2$. The resulting annual lake evaporation for the period 1966-2011 was about 1600 mm year$^{-1}$. The mean water balance components for the last period are shown in Table 6. When we used the water balance approach the computed evaporation varied much from month to month and year to year. This is an indication that some hydrological input data are uncertain. In any case, water balance computations over a long time period should give reasonable estimate of mean lake evaporation.





**Table 6. Monthly mean water balances for Lake Titicaca for the period 1966-2011.**

| Month | Ramis flow (m³/s) | Inflow (m³/s) | Outflow (m³/s) | Lake water depth (m) | Precipitation (mm/month) | Evaporation (mm/month) |
|---|---|---|---|---|---|---|
| Jan | 173.5 | 549.3 | 31.3 | 283.9 | 177.0 | 143.7 |
| Feb | 198.8 | 629.4 | 39.4 | 284.1 | 141.6 | 123.2 |
| Mar | 190.4 | 602.8 | 49.8 | 284.4 | 126.8 | 124.6 |
| Apr | 104.9 | 332.1 | 50.9 | 284.5 | 50.6 | 137.2 |
| May | 38.7 | 122.6 | 46.5 | 284.5 | 13.2 | 139.6 |
| Jun | 20.7 | 65.5 | 41.8 | 284.4 | 7.3 | 136.6 |
| Jul | 14.5 | 46.0 | 37.0 | 284.3 | 6.6 | 133.1 |
| Aug | 11.0 | 34.9 | 32.1 | 284.2 | 13.2 | 121.7 |
| Sep | 9.9 | 31.3 | 28.1 | 284.1 | 29.9 | 118.3 |
| Oct | 14.0 | 44.2 | 24.1 | 284.0 | 45.5 | 134.9 |
| Nov | 24.0 | 76.0 | 21.7 | 283.9 | 56.7 | 134.3 |
| Dec | 55.2 | 174.7 | 22.1 | 283.9 | 102.5 | 127.9 |

### 5.2 Using daily data

5    When the mass transfer approach is used, it is straight-forward to determine the daily evaporation. Using the energy balance, the change of heat storage in the lake must be determined with high resolution. Detailed water temperature measurements are not available, however. Instead, it was assumed that the temperature changes at a steady rate through individual months. August as an example since the temperature for the whole month changed very little ($0.2^{\circ}$C). The computed daily evaporation is shown for August 2015 in Fig. 10. There were two days with average wind exceeding 6 ms$^{-1}$. Consequently,

10    the evaporation was high during these days when the mass transfer approach was used.





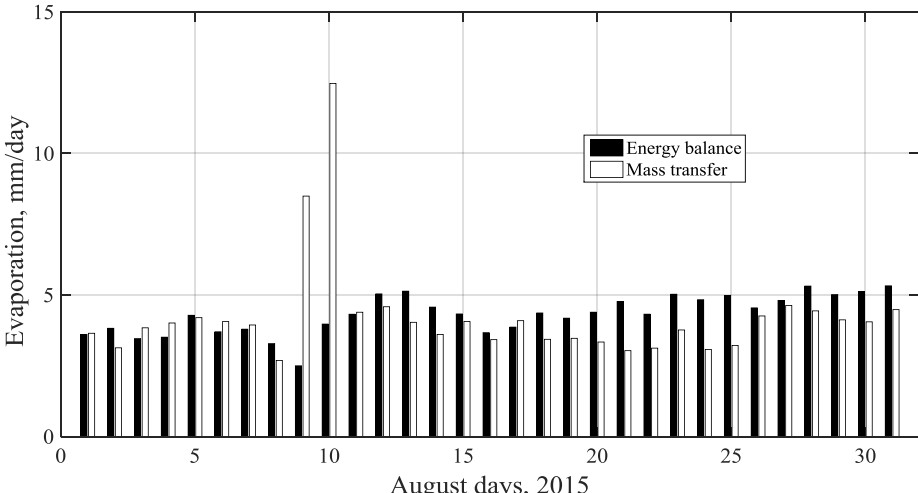

**Figure 10. Daily evaporation computed by the mass-transfer (shaded staples) and energy balance (filled staples) method.**

When annual evaporation was determined using daily data instead of monthly mean data, there was hardly any difference for the mass transfer method. As indicated above it is not possible to use the energy balance method with short time resolution when temperature changes from day to day have to be taken in to account. However, the evaporation can be computed neglecting the heat change keeping the Bowen ratio constant through-out a month and changing the Bowen ratio day by day. In this case, it was found that evaporation increased by about 2%. The conclusion, considering the many uncertainties involved in estimating evaporation, is that it is sufficient to use monthly means when estimating evaporation.

The evaporation computed with the Penman equation falls between what was found by the energy balance and the mass transfer approach being somewhat closer to the energy balance than to the mass transfer results. Since monthly means are sufficient for computing evaporation with the two above methods, mean values are sufficient also when using the Penman method.

**5.2.1 Evaporation for individual days**

When calculating evaporation using daily data, it was found that there are large differences between the methods. The maximum daily evaporation using the mass transfer method was 12 mm day[-1]. Neither the energy balance nor the Penman method gave higher evaporation than 8 mm day[-1]. There was a poor agreement between the mass transfer computed daily evaporation and the corresponding results using the other two methods.



## 6 Conclusions

Since information on river inflow to Lake Titicaca and precipitation on the lake surface are uncertain, it is suggested that the most reliable method of determining the annual lake evaporation is using the heat balance approach. To estimate the monthly lake evaporation using the heat balance, heat storage changes must be known. Since convection from the surface layer is

5    intense during nights resulting in a well-mixed top layer every morning, it is possible to determine the change of heat storage from the measured morning surface temperature. The lake evaporation is fairly uniformly distributed over the year with lows in July - September. The mean annual evaporation is about 1700 mm year$^{-1}$.

When using the mass transfer approach, the required coefficients in the aerodynamic equation was set so that the mean annual evaporation agreed with that obtained from the heat balance calculations. These coefficients were found to be lower

10   than coefficients used in previous studies. Also, when using the mass transfer approach, the evaporation was found to be lowest in July - September. Monthly evaporation computed using daily data and monthly means resulted in minor differences.

25

30



## 7 Acknowledgements

We would like to express our sincere appreciation to HASM, Research Programme – Hydrology of Altiplano from Space to Modeling, from Lab. GET-IRD and IHH-UMSA (Instituto de Hidráulica e Hidrología, UMSA, Bolivia), financed by TOSCA-CNES (Centre National d'Etudes Spatiales), especially to Franck Timouck from IRD/France, who lead the lake weather monitoring. We would like to thank Senamhi-Bolivia (Servicio Nacional de Hidrometeorología de Bolivia) for providing long-term climatic data. Thanks also to the IMARPE-Perú (Instituto para el Mar del Perú/Puno) for providing additional hydrological data and as well surface water temperature of Lake Titicaca. As well, our acknowledgment is directed to the project: Fortalecimiento de Planes Locales de Intervención y Adaptación al Cambio Climático en el Altiplano Boliviano at Agua Sustentable-Bolivia, for providing the Titicaca Lake discharge data. Finally, we thank the Programme Babel-Erasmus UE for providing economic assistance and completing this work in Sweden.

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
