# Peer review of "Modeling Lake Titicaca Daily and Monthly Evaporation"

_Hydrology and Earth System Sciences, 2018_

## Referee Comment (RC1) · Anonymous Referee #1 · 2 Jul 2018

This paper titled "Modeling Lake Titicaca Daily and Monthly Evaporation" basically deals with a critical subject that is of interest to many water scientists, namely evaporation in lakes. The authors comes up with comparing different methods to calculate daily and monthly evaporation from Lake Titicaca, to improve evaporation loss estimations. The idea and structure of this paper are clear, but there no innovative findings compared to previous studies in this paper. And I recommend this paper until the authors address some problems as follows.

Major comments: Abstracts: ïČŸThe abstract part should show the essence of the paper, including the significance of this research, methods used and conclusions. However, the authors paid too much attention to the research results while ignoring the data source and the significance of this paper. I suggest the authors add the content

[Figure]

I mentioned above in the abstract. Introduction: ïČŸThe introduction part is basically organized well. However, the methods or the models are ignored in this section, additional information on the theoretical background would be useful here. I suggest moving 3.1 section here. Methods: ïČŸIn this part, the authors pay too much attention to theoretical background, in my opinion, basic introduction and literature about the Theoretical background should be removed to introduction part. ïČŸFour evaporation estimation methods were applied in this study, water balance, energy balance, mass transfer, and the Penman method, I think the authors could add a reference for the equations. Conclusions: ïČŸIn the conclusion part, it would be more comprehensive and clear for the authors to conclude the significance as well as the limitation of the research, and with stating the limitations of this research, the suggested research direction for continued studies could be given at the end of this part. Minor comments: ïČŸPlease check the units throughout the paper. ïČŸPlease write the first occurrence of acronyms in full letters. ïČŸPlease check the references throughout the paper, the references exist in inconsistent or incomplete formats.

―――――――――――――――――――

---

## Referee Comment (RC2) · Anonymous Referee #2 · 16 Jul 2018

The authors present a study on assessing lake evaporation for Lake Titicaca. They compare different methods at the monthly time scale and then use the most successful methods to apply at the daily time scale. Honestly, although the paper is in principal OK (methodologically), I don't see much novelty in the paper. The authors apply different existing methods to a location that is measured before and even don't use any new method of analysing their results. So I don't see much added value in this study. For a journal like HESS, this should be the case.

specific comments:
-P2L13: why are daily observations/estimates necessary? It's not clear from the introduction. Please elaborate.

[Figure]

- after reading the introduction, I understand that Lake Titicaca is studied before with different methods. So why redo the experiment? Do you have any indication that the existing estimates are not OK?

-P6L16: So you don't trust the precipitation data on shore, so why don't you use e.g., remote sensing data? The lake is big enough, I would say.

-P8L27: disadvantage of this method is that a and b are empirical numbers. So you can question if these values found in Russia can be used in Lake Titicaca.

-P8eq11: the surface area A is a function of depth. I assume that the biggest error are caused by this.

-P8L20: I don't understand this sentence. Why is daily evaporation not important for the water balance? you can apply the water balance at any time scale you want.

-P14section 5.1: be consistent with the naming of your methods. Now the method 'carmouze' is used, while before it was named mass transfer method. This is confusing for the reader.

-P14fig 5: how can you compare evaporation data of two different years? Would be weird if they were the same.

-P16L10: I think the biggest error is not the water level, but the associated wrong estimation of the surface area...

-P19L6:?? are you keeping the bowen ratio constant of do you change it day by day? Confusing sentence. Please rewrite.

minor/technical:
-P1L19: ".. using THE heat balance.."
-P1L22: unit of annual evaporation is mm/YEAR
-P3L6,7,8,9,10,11: '-1' should be superscript
-P3L12: unit of annual evaporation is mm/YEAR
-P5L16-18: unit of annual evaporation is mm/YEAR
-P5L24-25: celsius degree symbol is not ok
-P7section3.2.1: add units to all variables.

-P8eq6: what is P? not defined.
-P8L25-26: Please correct this sentence. It's double.
-P8L27: please specify units and avoid 'with units as above".
-P9section 3.2.3: Please provide units and don't use "With units as above".
-P14L5: remove 'a total evaporation of 3400 mm, corresponding to". It's not important.
-P14L12:unit of annual evaporation is mm/YEAR
-P16L9:unit of annual evaporation is mm/YEAR

―――――――――――――――

---

## Short Comment (SC1) · 29 Jul 2018

**1-Modeling Lake Titicaca Daily and Monthly Evaporation**

**Dear reviewer:**

Surprisingly the climate warming in this regions today exceeds the average global warming, the evaporation is one the variables that might be altered enormously. Then as it is pointed out the evaporation issue is crucial for this lake, also the interest is to propose a practical models for their implementation, beside the couple of models proposed already by Delclaux et al. (2006), which are dependent on only solar radiation and wind factor data.

According to the previous studies of evaporation in other places there have been found important differences by applying the daily and monthly computation. We think that in order to study the climate change assessment on the models at different time scale, for the case of Titicaca Lake, must be defined the appropriate models at both time scales and also based on our available data. For the first time we obtained high resolution met data for this Lake in the last couple of years. Thus the outputs from climate changes scenarios at any time scale should be able to analysis for this Lake.

**Major comments:**

C1: The abstract part should show the essence of the paper, including the significance of this research, methods used and conclusions. However, the authors paid too much attention to the research results while ignoring the data source and the significance of this paper I suggest the authors add the content I mentioned above in the abstract

Answer to comment-C1:
- Yes we will include the recommendations about the significance of this research and as well about the new data used on this paper. Yes the interest of the paper is not only the results, mainly are the models defined for the climate change assessing, on which context with the discussions about the possibility of using, and finally is to show the results and between different time scales.

As so far we have rewritten the abstract of the paper as it follows:

- **Abstract.** Lake Titicaca is a crucial water resource for the Altiplano, in the central part of the Andean Mountain range, and one of the lakes most affected by climate warming. Since surface evaporation explains most of the lake´s water losses, reliable estimates are paramount for the prediction of global warming impacts on the Lake Titicaca and for the region´s water resources planning and adaptation to climate changes. This study investigated the suitability of fours methods for the assessment of Lake Titicaca´s evaporation at daily and monthly time scales. These methods are: water balance, heat balance, mass transfer and Penman´s equation. Evaporation losses were calculated following the four methods using both, daily meteorological records and their monthly averages. We found that the most reliable method for determining the annual lake evaporation was the heat balance approach, although the Penman equation allows an easier implementation based on generally available meteorological parameters. The main difficulty for the use

of the heat balance method is that heat storage changes must be knowing in advance. Since convection from the surface layers is intense during nights resulting in well-mixed top layer every morning, changes in heat storage were estimates from the measured morning surface temperature. The mean annual lake evaporation was for to be1700 mm year$^{-1}$. Monthly evaporation computed using daily and monthly mean between the models results in minor differences.

C2: The introduction part is basically organized well. However, the methods or the models are ignored in this section, additional information on the theoretical background would be useful here. I suggest moving 3.1 section here

Answer to comment-C2:
- Yes is a good idea to move to this chapter the section 3.1, also beside the theoretical improvement the paper objective also will be refined based on the above background.

C3: Method: In this part, the authors pay too much attention to theoretical background, in my opinion, basic introduction and literature about the Theoretical background should be removed to introduction part

Answer to comment-C3:
- Yes some of the theoretical background will be removed or moved to the upper. Yes the references to the equations will be included.

C4: Four evaporation estimation methods were applied in this study, water balance, energy balance, mass transfer, and the Penman method, I think the authors could add a reference for the equations.

Answer to comment C4:
- Yes the authors of all the equation will added.

C5: In the conclusion part, it would be more comprehensive and clear for the authors to conclude the significance as well as the limitation of the research, and with stating the limitations of this research, the suggested research direction for continued studies could be given at the end of this part.

Answer to comment-5: Conclusions
- About the limitations and the advantages about the models used we will describe, especially in respect to models to be used for climate changes assessing at different time scales. Then will have the significance of the study, but also pointing out some recommendations that the models used might improve it in the future using spatial observation data for instance.

**Minor comments:**

Answer to comment-5: Final comments.
- Yes, we realized that units must be included, also we will correct the symbols and the reference chapter.

An advance, thank you very much for your valuable remarks for improving our paper.

---

## Short Comment (SC2) · 29 Jul 2018

**2-Modeling Lake Titicaca Daily and Monthly Evaporation**

**Dear reviewer:**

Surprisingly the climate warming in this regions today exceeds the average global warming, the evaporation is one the variables that might be altered enormously. Then as it is pointed out the evaporation issue is crucial for this lake, also the interest is to propose a practical models for their implementation, beside the couple of models proposed already by Delclaux et al. (2006), which are dependent on only solar radiation and wind factor data.

According to the previous studies of evaporation in other places there have been found important differences by applying the daily and monthly computation. We think that in order to study the climate change assessment on the models at different time scale, for the case of Titicaca Lake, must be defined the appropriate models at both time scales and also based on our available data. For the first time we obtained high resolution met data for this Lake in the last couple of years. Thus the outputs from climate changes scenarios at any time scale should be able to analysis for this Lake.

As so far we have rewritten the abstract of the paper as it follows:

- **Abstract.** Lake Titicaca is a crucial water resource for the Altiplano, in the central part of the Andean Mountain range, and one of the lakes most affected by climate warming. Since surface evaporation explains most of the lake´s water losses, reliable estimates are paramount for the prediction of global warming impacts on the Lake Titicaca and for the region´s water resources planning and adaptation to climate changes. This study investigated the suitability of fours methods for the assessment of Lake Titicaca´s evaporation at daily and monthly time scales. These methods are: water balance, heat balance, mass transfer and Penman´s equation. Evaporation losses were calculated following the four methods using both, daily meteorological records and their monthly averages. We found that the most reliable method for determining the annual lake evaporation was the heat balance approach, although the Penman equation allows an easier implementation based on generally available meteorological parameters. The main difficulty for the use of the heat balance method is that heat storage changes must be knowing in advance. Since convection from the surface layers is intense during nights resulting in well-mixed top layer every morning, changes in heat storage were estimates from the measured morning surface temperature. The mean annual lake evaporation was for to be1700 mm year$^{-1}$. Monthly evaporation computed using daily and monthly mean between the models results in minor differences.

**Specific comments:**

P2L13: why are daily observations/estimates necessary? It's not clear from the introduction. Please elaborate.

Answer to P2L13:

- Is correct in the Introduction/objective, abstract and in the discussions chapters it was not highlighted in respect to daily evaporation computation. In fact today is necessary to have the models at time scale for testing climate change scenarios outputs. According the previous studies of evaporation (at daily and month scales) depending on model scale used the results obtained might differentiated as well. This aspect will be elaborate in the introduction part definitely.

  Yes, as it was mentioned in above many empirical models were used at month scale for this lake, but still we have the curiosity on computing at daily scale since for first time we have access to high resolution data, second in order to test the climate changes scenarios we will need to have the models at this scale as well. Finally, we think that the results of evaporation might improve from this perspective.

P6L16: So you don't trust the precipitation data on shore, so why don't you use e.g., remote sensing data? The lake is big enough, I would say.

Answer to P6L16:
- Regarding the quality precipitation data for the Lake we have considered two rain gauges stations; thus it not might very representative for the entire surface area. Farther more the precipitation could the most uncertainty data, in particular in this region because the long-time period it was measured manually. It seems very good idea to use from remote sensing at least for the two research years (2015-2016), we will compute on this way.

P8L27: disadvantage of this method is that a and b are empirical numbers. So you can question if these values found in Russia can be used in Lake Titicaca.

Answer to P8L27:
- Yes, the a (mm mbar$^{-1}$ day$^{-1}$) and b (mm mbar$^{-1}$ s m$^{-1}$) parameters are empirical values in the mass transfer equation, and by using the previous values found by Carmouze, the evaporation rate was higher to the rest, even significantly higher. Then we lowered those parameters substantially (from a=0.7 and b=0.30 up to a=0.17 and b=0.155). Definitely, Russian values does not mean that can be used for our case, also Russian values cannot be the minimum limits. Since the actual values found almost is the average between the other existing values maybe we do not need to be redundant.

P8eq11: the surface area A is a function of depth. I assume that the biggest error are caused by this.
Answer to P8eq.11:
- Yes A=f(h), as it was anticipated we used just for A as average value, but since the computation will test by precipitation derived from satellite source, we are able to verify this problem.

P8L20: I don't understand this sentence. Why is daily evaporation not important for the water balance? you can apply the water balance at any time scale you want.

Answer to P8L20:
- Yes it is correct, we can compute at any time scale the water balance, in case of Titicaca Lake due its size the daily evaporation value should affect very little on

the water balance; however the monthly or yearly values might define the lake status. We will analyze more the thinking on the text. However we want to point out that monthly balance modeling is crucial for everything; thus its analysis and accuracy as well.

P14section 5.1: be consistent with the naming of your methods. Now the method 'carmouze' is used, while before it was named mass transfer method. This is confusing for the reader.

Answer to P14section 5.1.
- Ok, the redaction is very easy to correct here.

-P14fig 5: how can you compare evaporation data of two different years? Would be weird if they were the same.

Answer to P14fig 5:
- In the paper we analyzed the full continuous met data gathered during 2015 and 2016 and according to consistent flied campaign. By other hand we could not obtain a good experimental evaporation data (tank evaporation data). I fact we are comparing the methods for each year because we need to be sure that methods and the data work, especially for the monthly values like was anticipated. In the same way that the radiative parameters were compared. All the authors have been show until now only the mean month values or yearly.

P16L10: I think the biggest error is not the water level, but the associated wrong estimation of the surface area...

Answer to P16L10:
- This problem will be correct with new computation of water balance as mentioned already.

-P19L6:?? are you keeping the bowen ratio constant of do you change it day by day? Confusing sentence. Please rewrite.

Answer to P19L6:
- The Bowen ratio changes day by day since we have the observed data at daily.

**Minor comments:**

-P1L19: ".. using THE heat balance.."
-P1L22: unit of annual evaporation is mm/YEAR
-P3L6,7,8,9,10,11: '-1' should be superscript
-P3L12: unit of annual evaporation is mm/YEAR
-P5L16-18: unit of annual evaporation is mm/YEAR
-P5L24-25: celsius degree symbol is not ok
-P7section3.2.1: add units to all variables.

Answer to minor comments:

- In this regard all the remarks already have been corrected.

---

## Referee Comment (RC3) · Anonymous Referee #1 · 30 Jul 2018

I have read the revised manuscript and they have revised and covered the all of my initial comments. From my point of view I have no further comments. I think it can be accepted for publication at the current form.

---

## Short Comment (SC3) · 30 Jul 2018

The authors estimate evaporation from Titikaka Lake which is the main fresh water body in Bolivia and Peru, and it is vital for the arid Altiplano region. Indeed, the results will be very useful for future water resources management in the Altiplano, which is a dry and vulnerable area.

The article is clear with come comments already mentioned by previous comments and properly addressed by authors. I would like to add some suggestions.

*Authors collected data from different locations covering a wide area. I get the doubt about the spatial variability of the evaporation. Some year ago, I analyzed spatial variation of evaporation in part of the Bolivian Altiplano using remote sensing data from

MOD16 (Moya Quiroga et al., 2014). Unfortunately, MOD16 does not provide data for the Titikaka lake because of passive imagery limitations in the Titikaka lake and water bodies. Therefore, it would be interesting if the authors provide a map (and discussion) showing the spatial variability of the evaporation. This spatial variable evaporation would also an important novelty as requested by reviewers.

*Authors collected data from the hydrological year 2015-2016. Climatological conditions in the bolivian Altiplano are highly influenced by El Niño South Oscillation (ENSO). I believe it would be important to provide some information and discussion about the climate conditions on that year. Was it Niño or Niña?

Reference

-Moya Quiroga V., Mano A., Asaoka Y., Udo K., Kure S., and Mendoza J.: Evaluacion de la evapotranspiracion potencial estimada mediante sensores remotos de la mision MODIS: La cuenca Condoriri del Altiplano Boliviano, XXVI Congreso Latinoamericano de Hidraulica, Santiago, Chile, 2014.

---

## Author Comment (AC1) · 27 Aug 2018

Dear reviewer,

Please, find attached the new manuscript with all the corrections.

Many thanks in advance for your collaboration.

Best, Ramiro

Please also note the supplement to this comment:
https://www.hydrol-earth-syst-sci-discuss.net/hess-2018-127/hess-2018-127-AC1-supplement.pdf

---

## Author Comment (AC4) · 29 Aug 2018

**1-Modeling Lake Titicaca Daily and Monthly Evaporation**

**Dear reviewer:**

Surprisingly the climate warming in this regions today exceeds the average global warming, the evaporation today is one the variables that might be altered enormously. Then as it is pointed out the evaporation issue is crucial for this lake, also the interest is to propose a practical models for their implementation, beside the couple of models proposed already by Delclaux et al. (2006), which are dependent on limited variables, like solar radiation and wind factor data.

According to the previous studies of evaporation in other places there have been found important differences by applying the daily and monthly computation. We think that in order to study the climate change assessment through the models at different time scale, for the case of Titicaca Lake, must be defined the appropriate models at both time scales, and also based on our available data. For the first time we obtained high resolution met data for this Lake in the last couple of years. Thus the outputs from climate changes scenarios at any time scale should be able to analysis for this Lake.

**Major comments:**

As so far we have rewritten the abstract of the paper as it follows:

**Abstract.** Lake Titicaca is a crucial water resource in the central part of the Andean Mountain range, which is one of the lakes most affected by climate warming. Since surface evaporation explains most of the lake´s water losses, reliable estimates are paramount for the prediction of global warming impacts on Lake Titicaca and for the region´s water resources planning and adaptation to climate change. Evaporation estimates were done in the past at monthly time steps and using the four methods, as follows: water balance, heat balance, and mass transfer and Penman´s equation. The obtained annual evaporation values showed significant dispersion. This study used new, daily frequency hydro-meteorological measurements. Evaporation losses were calculated following the mentioned methods using both, daily records and their monthly averages, to assess the impact of higher temporal resolution data in the evaporation estimates. Changes in the lake heat storage needed for the heat balance method were estimated based on the morning water surface temperature, because convection during nights results in a well-mixed top layer every morning over a constant temperature depth. We found that the most reliable method for determining the annual lake evaporation was the heat balance approach, although the Penman equation allows an easier implementation based on generally available meteorological parameters. The mean annual lake evaporation was found to be 1700 mm year$^{-1}$. This value is considered an upper limit of the annual evaporation since the main study period was abnormally warm. The obtained upper limit lowers by 200 mm year$^{-1}$ the highest evaporation estimation obtained previously, thus reducing the uncertainty in the actual value. Regarding the evaporation estimates using daily and monthly averages, these resulted in minor differences for all methodologies.

**Comment 1:**

Some years ago, I analyzed spatial variation of evaporation in part of the Bolivian Altiplano using remote sensing from MOD16. Unfortunately, MOD16 does not provide data for the Titicaca Lake because of passive imagery limitation. Therefore, it would be interesting if the authors might provide the spatial variability of the evaporation.

**Answer to comment 1:**

Within the territory of Bolivia are established the three main watershed: Altiplano (or TDPS) La-Plata and Amazon, several data source of direct measurement (e.g., radar or satellite) were used recently, mostly for the rainfall spatial analysis (Heidinger et al., 2012; Moya, et al., 2014, Blacutt et al., 2015; Satge et al., 2015, 2017) or for comparison. All the resources showed some discrepancy when using different rainfall features over the regions, for instance, TMPA showed more accurate for the Amazon an for the TDPS, while IMERG of the La-Plata (Satge et al., 2017). Moreover all the product during the rainy season between December-March might increase the performance and less for the dry season. Furthermore Lake Titicaca with extensive water surface area close to 9000 $km^2$, and here some of SREs estimated deficit rainfall, explained and probably due to emissivity contrast, which are: TMPA-3B42, PERSIANN and GSMaP. Also we understood that on MOD16 also there as found similar problems by the time you applied remote sensing data.

In the next paper exactly we will be focused on the spatial evaporation analysis and probably taken into account the MOD data, with more terrain observed data since we have the predicting models. On this aspect your concern really is well focused.

**Comment 2**:

Authors collected data from the hydrological years 2015-2016. Climatological conditions in the Altiplano are highly influenced by El-Niño South Oscillation (ENSO). I believe it would be important to provide some information and discuss about the climate conditions on that year.

**Answer to comment 2:**

We are totally agree with you, part of 2015 and 2016 was one of the strongest El-Niño in the Altiplano, even though the world (http://www.ciifen.org). For sure we will analyze our results related to that event. For that, we will analyze more the air temperature data for longer period; then we should be able to detect the influence on our results the warmer period taken in those months.

In advance, that you very much for your valuable contribution for improving our research.

All the best,

Ramiro

---

## Author Comment (AC5) · 5 Sep 2018

Dear Reviewer, One more time many thanks for your reply. The last attached manuscript includes all the remarks sent us. Especially the water balance for short period was recalculated, including the derivated statistics, on the same time all the water balance variables one more time were analyzed, where the most problematic variables might the lake discharge data. Please, let us to know if any extra comments do you have for us. All the best, Ramiro

---

## Author Comment (AC6) · 18 Sep 2018

Dear Reviewer,

Please, find attached the last version manuscript, which includes the two reviewer's comments. The manuscript one more time was rewritten. Please, let us to know any further comments. Thank you very much,

Ramiro

Please also note the supplement to this comment:
https://www.hydrol-earth-syst-sci-discuss.net/hess-2018-127/hess-2018-127-AC6-supplement.pdf

---

## Author Response (AR1)

**Modeling Lake Titicaca Daily and Monthly Evaporation**

R. Pillco Zolá[1], L. Bengtsson[2], R. Berndtsson[2], B. Martí-Cardona[3], F. Satgé[4], F. Timouk[5], M.-P. Bonnet[6], L. Mollericon[1], C. Canedo[1], C. Gamarra[7], J. Pasapera[7],

[1] Instituto de Hidráulica e Hidrología, Universidad Mayor de San Andrés, La Paz, Bolivia
[2] Division of Water Resources Engineering and Center for Middle Eastern Studies, Lund University, Lund, Sweden
[3] Department of Civil and Environmental Eng., University of Surrey, Guildford, UK
[4] CNES, UMR HydroSciences, Univeristy of Montpellier, Place E. Bataillon, 34395 Montpellier Cedex 5, France
[5] IRD, UMR5563); Obs. Midi-Pyrénées, Université P. Sabatier, Toulouse, France
[6] IRD, UMR Espace-Dev, Maison de la télédétection, 500 rue JF Breton, 34093 Montpellier cedex 5, France
[7] IMARPE, Puno, Perú

*Correspondence to*: R. Pillco Zolá (rpillco@umsa.edu.bo)

**Answer to Referee 1:**

C1: The abstract part should show the essence of the paper, including the significance of this research, methods used and conclusions. However, the authors paid too much attention to the research results while ignoring the data source and the significance of this paper I suggest the authors add the content I mentioned above in the abstract

Answer to comment-C1:
- Yes we will include the recommendations about the significance of this research and as well about the new data used on this paper. Yes the interest of the paper is not only the results, mainly are the models defined for the climate change assessing, on which context with the discussions about the possibility of using, and finally is to show the results and between different time scales.

**New abstract**

**Abstract** Lake Titicaca is a crucial water resource in the central part of the Andean Mountain range, which is one of the lakes most affected by climate warming. Since surface evaporation explains most of the lake´s water losses, reliable estimates are paramount for the prediction of global warming impacts on Lake Titicaca and for the region´s water resources planning and adaptation to climate change. Evaporation estimates were done in the past at monthly time steps and using the four methods, as follows: water balance, heat balance, and mass transfer and Penman´s equation. The obtained annual evaporation values showed significant dispersion. This study used new, daily frequency hydro-meteorological measurements. Evaporation losses were calculated following the mentioned methods using both, daily records and their monthly averages, to assess the impact of higher temporal resolution data in the evaporation estimates. Changes in the lake heat storage needed for the heat balance method were estimated based on the morning water surface temperature, because convection during nights results in a well-mixed top layer every morning over a constant temperature depth. We found that the most reliable method for determining the annual lake evaporation was the heat balance approach, although the Penman equation allows an easier implementation based on generally available meteorological parameters. The mean annual lake evaporation was found to be 1700 mm year$^{-1}$. This value is considered an upper limit of the annual evaporation since the main study period was abnormally warm. The obtained upper limit lowers by 200 mm year$^{-1}$ the highest evaporation estimation obtained previously, thus reducing the uncertainty

in the actual value. Regarding the evaporation estimates using daily and monthly averages, these resulted in minor differences for all methodologies.

Key words: Lake Titicaca, heat balance, water balance, lake evaporation.

C2: The introduction part is basically organized well. However, the methods or the models are ignored in this section, additional information on the theoretical background would be useful here. I suggest moving 3.1 section here

Answer to comment-C2:
- Yes is a good idea to move to this chapter the section 3.1, also beside the theoretical improvement the paper objective also will be refined based on the above background.

**1 Introduction**

[revised manuscript text omitted]

C3: Method: In this part, the authors pay too much attention to theoretical background, in my opinion, basic introduction and literature about the Theoretical background should be removed to introduction part

Answer to comment-C3:
- Yes some of the theoretical background will be removed or moved to the upper. Yes the references to the equations will be included.

C4: Four evaporation estimation methods were applied in this study, water balance, energy balance, mass transfer, and the Penman method, I think the authors could add a reference for the equations.

Answer to comment C4:
- Yes the authors of all the equation will added.

**On the following section (3 Methods) were answered to comments C3 and C4 and Minor comments:**

Answer to comment-5: Final comments.
- Yes, we realized that units must be included, also we will correct the symbols and the reference chapter.

[revised manuscript text omitted]

C5: In the conclusion part, it would be more comprehensive and clear for the authors to conclude the significance as well as the limitation of the research, and with stating the limitations of this research, the suggested research direction for continued studies could be given at the end of this part.

Answer to comment-5: Conclusions
- About the limitations and the advantages about the models used we will describe, especially in respect to models to be used for climate changes assessing at different time scales. Then will have the significance of the study, but also pointing out some recommendations that the models used might improve it in the future using spatial observation data for instance.

**6 Conclusions**

Due to uncertainty of most observed data such as river inflow to Lake Titicaca, and mainly the discharge data, it might be no easy to improve the water balance results; then it is suggested that the most reliable method of determining the lake evaporation is using the heat balance approach. To estimate the lake evaporation using this method, heat storage changes must be known. Since convection from the surface layer is intense during nights resulting in a well-mixed top layer every morning, it is possible to determine the change of heat storage from the measured morning surface temperature. The lake evaporation is fairly uniformly distributed over the year with lows between July and September. The mean annual evaporation is about 1700 mm year$^{-1}$, and the mean monthly

evaporation is 141.8 mm month$^{-1}$. When using the mass transfer approach, the required coefficients in the aerodynamic equation was set so that the mean annual evaporation agreed with that obtained from the heat balance calculations. These coefficients were found to be lower than coefficients used in previous studies. Also, when using the mass transfer approach, the evaporation was found to be lowest in July - September.

However, for the climate changes effect on Titicaca Lake evaporation assessment purposes the practical approach rather than the two empirical models might be the Penman equation, one due to available observed data for this lake, and two due to integral behavior of the equation. Also in comparison with the two models proposed in Delclaux et al. (2007) for modeling the lake evaporation, where the first model only depends on the solar radiation data, and the second one depends plus on the air temperature factor; thus both models cannot be applied broadly. The Penman model based on the adjusted wind coefficient, the mean annual evaporation is 1620 mm year$^{-1}$ and the mean monthly is 135 mm month$^{-1}$. As so far, monthly evaporation computed using daily data and monthly means resulted in minor differences. The most practical model for using at daily scale might be the mass transfer approach and the Penman in comparison to energy balance approach for being high demand observed data. Particularly the Penman equation at daily temporal scale correctly might applied for the climate changes assessment at this altitude. Nonetheless, according to spatial available data from remote sensing, the evaporation equations used at daily and month scales could be applied from now for improving the spatial pattern of the lake evaporation. Since we had really extreme single warmer days during the period 2015-2016 due to El-Niño phenomenon, must be expected to have higher daily rates of evaporation; therefore the application of the models at both time scales for the study period we believe that was found the upper limits of yearly evaporation.

**Minor comments:**

Answer to comment-5: Final comments.
- Yes, we realized that units must be included, also we will correct the symbols and the reference chapter.

**References**

[revised manuscript text omitted]

**Answer to Referee 2:**

**Specific comments:**

P2L13: why are daily observations/estimates necessary? It's not clear from the introduction. Please elaborate.

Answer to P2L13:

- Is correct in the Introduction/objective, abstract and in the discussions chapters it was not highlighted in respect to daily evaporation computation. In fact today is necessary to have the models at time scale for testing climate change scenarios outputs. According the previous studies of evaporation (at daily and month scales) depending on model scale used the results obtained might differentiated as well. This aspect will be elaborate in the introduction part definitely.

  Yes, as it was mentioned in above many empirical models were used at month scale for this lake, but still we have the curiosity on computing at daily scale since for first time we have access to high resolution data, second in order to test the climate changes scenarios we will need to have the models at this scale as well. Finally, we think that the results of evaporation might improve from this perspective.

[revised manuscript text omitted]

**P6L16**: So you don't trust the precipitation data on shore, so why don't you use e.g., remote sensing data? The lake is big enough, I would say.

Answer to P6L16:
- Regarding the quality precipitation data for the Lake we have considered two rain gauges stations; thus it not might very representative for the entire surface area. Farther more the precipitation could the most uncertainty data, in particular in this region because the long-time period it was measured manually. It seems very good idea to use from remote sensing at least for the two research years (2015-2016), we will compute on this way.

**Old versión of Figure 9**

[Figure]

**Figure 1. Monthly actual evaporation calculated by the four methods for the period January 2015 to December 2016.**

**New version of Figure 9**

[Figure]

**Figure 2. Monthly actual evaporation calculated by the four methods for the period January 2015 to December 2016.**

In the Figure 9, the precipitation data the most controversial, it was analysed from other point and neighbour observation, also placed close to the lake shore. Thus the water balance evaporation now gave us a bit different from the previous ones, and it fits better with other three models. For this computation also was analysed the Q-out form the lake; nonetheless some Q data measured for the period 2015-2916 still might be suspicious. The changes of surface water according to water levels dynamics now was taken into the computation, since we have the relation established A=f(h) for the lake. To use the remote sensing data by is considered to be included in another paper; also from the previous works the images used for derivation precipitation data is not necessarily accurate for this lake due to high emissivity.

P8L27: disadvantage of this method is that a and b are empirical numbers. So you can question if these values found in Russia can be used in Lake Titicaca.

Answer to P8L27:
- Yes, the a (mm mbar$^{-1}$ day$^{-1}$) and b (mm mbar$^{-1}$ s m$^{-1}$) parameters are empirical values in the mass transfer equation, and by using the previous values found by Carmouze, the evaporation rate was higher to the rest, even significantly higher. Then we lowered those parameters substantially (from a=0.7 and b=0.30 up to a=0.17 and b=0.155). Definitely, Russian values does not mean that can be used for our case, also Russian values cannot be the minimum limits. Since the actual values found almost is the average between the other existing values maybe we do not need to be redundant.

**Mass transfer approach**

The mass-transfer aerodynamic approach is used in various models based on Dalton's law (Dalton, 1802). The latent heat transfer is related to the vapor pressure deficit. Most often a linear wind function is used (e.g., Carmouze et al., 1992)

$$E = (a + bU\ )(e_w - e_a)$$
(9)

Where $E$ is evaporation rate, $U$ is wind velocity (m s$^{-1}$), ($e_w$ - $e_a$) is vapor pressure deficit (mbar). The parameter $a$ accounts for unstable atmospheric conditions. Carmouze $et\ al.$ (1992) used $a$=0.17 (mm mbar$^{-1}$ day$^{-1}$) and $b$=0.30 (mm mbar$^{-1}$ s m$^{-1}$).

P8eq11: the surface area A is a function of depth. I assume that the biggest error are caused by this.
Answer to P8eq.11:
- Yes A=f(h), as it was anticipated we used just for A as average value, but since the computation will test by precipitation derived from satellite source, we are able to verify this problem.

The changes of surface water according to water levels dynamics now was taken into the computation, since we have the relation established A=f(h) for the lake

P9L20: I don't understand this sentence. Why is daily evaporation not important for the water balance? you can apply the water balance at any time scale you want.

Answer to P8L20:
- Yes it is correct, we can compute at any time scale the water balance, in case of Titicaca Lake due its size the daily evaporation value should affect very little on the water

balance; however the monthly or yearly values might define the lake status. We will analyze more the thinking on the text. However we want to point out that monthly balance modeling is crucial for everything; thus its analysis and accuracy as well.

The evaporation during individual days is not important for the water balance but only over longer periods as months. However, since the equations for calculating evaporation are not linear, the monthly evaporation computed from monthly mean meteorological data may differ from what is found when data with higher time resolution are used. In the aerodynamic approach the wind speed is multiplied by the vapor deficit. The energy balance approach includes the Bowen ratio, which may differ from day to day and can even be negative for certain periods. If high atmospheric vapor pressure is related to strong winds, the aerodynamic equation using monthly means can yield lower evaporation estimates than when daily values are used. This is further discussed below. The Bowen ratio changes during a month. When the net radiation is large, the air temperature is likely to be rather high but not necessarily related to high vapor pressure. For this situation, the Bowen ratio is relatively high and the computed evaporation higher than it would have been using a constant monthly Bowen ratio. This means that using monthly averages, the computed evaporation will tend to be low.

Yes, the last paragraph was analyzed one more time; thus there was not finding any contradiction.

P14section 5.1: be consistent with the naming of your methods. Now the method 'carmouze' is used, while before it was named mass transfer method. This is confusing for the reader.

Answer to P14section 5.1.
  -   Ok, the redaction is very easy to correct here.

Detailed energy balance computations over the period 2015-2016 should give good estimates of the total lake evaporation for that period. After 24 months the lake surface temperature at Puno more or less returned to the temperature at the beginning of 2015. Applying this method over the two years of study the mean annual lake evaporation is 1700 mm year$^{-1}$. When computing the evaporation month by month the change of heat storage was considered in the way previously described. The mixing depth was set to 40 m. The change of the heat storage is shown in Fig. 5. The values suggested by Carmouze et al. (1992) are shown for comparison. The calculated monthly heat storage agrees well with the Carmouze estimates.

[Figure]

**Figure 3. Change of heat storage during 2015-2016.**

-P14fig 5: how can you compare evaporation data of two different years? Would be weird if they were the same.

Answer to P14fig 5:
- In the paper we analyzed the full continuous met data gathered during 2015 and 2016 and according to consistent field campaign. By other hand we could not obtain a good experimental evaporation data (tank evaporation data). I fact we are comparing the methods for each year because we need to be sure that methods and the data work, especially for the monthly values like was anticipated. In the same way that the radiative parameters were compared. All the authors have been show until now only the mean month values or yearly.

P16L10: I think the biggest error is not the water level, but the associated wrong estimation of the surface area...

Answer to P16L10:
- This problem will be correct with new computation of water balance as mentioned already.
The new results from the computation is shown in above (Figure 9).

P19L6:?? are you keeping the bowen ratio constant of do you change it day by day? Confusing sentence. Please rewrite.

Answer to P19L6:
- The Bowen ratio changes day by day since we have the observed data at daily.

**Minor comments:**

-P1L19: ".. using THE heat balance.."
-P1L22: unit of annual evaporation is mm/YEAR
-P3L6,7,8,9,10,11: '-1' should be superscript
-P3L12: unit of annual evaporation is mm/YEAR
-P5L16-18: unit of annual evaporation is mm/YEAR
-P5L24-25: celsius degree symbol is not ok
-P7section3.2.1: add units to all variables.

Answer to minor comments:

- In this regard all the minor comments already have been added in the new manuscript.